# Research on state perception of scraper conveyor based on one-dimensional convolutional neural network

Jie Lu[1,2☉¤]*, Zhenlin Liu[1☉], Chenhui Han[1], Zhiqiang Yang[3], Jialu Zheng[1], Wangjie Zhang[1]

1 School of Coal Engineering, Datong University Shanxi Province, Datong, China, 2 School of Mines, China University of Mining and Technology, Xuzhou, China, 3 Zhongtian He Chuang Energy Co., LTD, Eedos, China

☉ These authors contributed equally to this work.
¤ Current address: Department of Mining Engineering, Shanxi Datong University, Datong, Shanxi Province, China
* lujie0114@cumt.edu.cn

**Data Availability Statement:** All relevant data are within the manuscript and its Supporting Information files.

## Abstract

Addressing the challenges of current scraper conveyor health assessments being influenced by expert knowledge and the relative difficulty in establishing degradation models for equipment, this study proposed a method for assessing the health status of scraper conveyors based on one-dimensional convolutional neural networks (1DCNN). The approach utilizes four preprocessed monitoring signals representing different health states of the scraper conveyor as input sources. Through multiple transformations of the data using a constructed one-dimensional convolutional neural network model, it extracts effective features from the data and establishes a mapping relationship between input data and equipment health status. This enables the recognition of the health status of the scraper conveyor. Comparative experimental analysis indicates that the proposed method can effectively identify the health status of the scraper conveyor, achieving an accuracy rate of 98.9%. This method provides an effective means and technical support for the subsequent health management of scraper conveyors in coal mining fully mechanized workfaces.

## Introduction

The scraper conveyor is currently the only transportation equipment used in fully mechanized coal mining workfaces. Its safe and stable operation is a crucial factor for reliable production in these workfaces. A shutdown of the scraper conveyor can disrupt the entire production chain in the mine, leading to incalculable losses [1–3]. Therefore, strengthening the monitoring of various parts of the scraper conveyor, efficiently analyzing monitoring data, and real-time perception of equipment health status are essential. These measures play an extremely important role in ensuring the stable and reliable operation of the scraper conveyor, improving equipment efficiency, and extending equipment lifespan.

**Funding:** This work was supported by the Research Funds for 2021 Shanxi Datong University (No. 2021K9.) and the Shanxi Datong University Graduate Education Innovation Project (No. 22CX45) and the Graduate Education Innovation Project of Shanxi Datong University in 2022 (22CX43) and the Graduate Education Innovation Project of Shanxi Datong University in 2022 (23CX54).

**Competing interests:** The authors have declared that no competing interests exist.

Currently, methods for assessing the health status of electromechanical equipment fall into three categories: data-driven health status assessment, knowledge-driven health status assessment, and model-driven health status assessment [4]. Knowledge-driven health assessment methods leverage expert knowledge and experience for equipment health status assessment, commonly using methods such as entropy weighting, subjective weighting, fuzzy comprehensive judgment, etc. Model-driven health assessment involves assessing the condition of a device or component by constructing physical mathematical models, with methods including the Kalman filtering algorithm, system and dynamic models, fault tree analysis, etc. Data-driven health status assessment utilizes sensor technology and data analysis to deeply analyze various monitoring data of mechanical equipment, determining the health status of the equipment. Commonly used data-driven health assessment methods include Bayesian networks [5], support vector machines [6], Markov theory [7], neural networks [8–10], etc. The types of health status assessment methods and their commonly used methods are illustrated in Fig 1.

Among the three mentioned methods, knowledge-driven health status assessment methods are susceptible to limitations imposed by existing expert knowledge and experience. There are subjective issues in determining the weights of various influencing factors, affecting the assessment results of equipment health status. Model-driven health assessment faces challenges in accurately representing the non-linear degradation characteristics of equipment using mathematical models, and the complexity of model establishment. With the maturation of various monitoring technologies, data-driven health status assessment, compared to establishing equipment degradation models, is simpler. The model can adaptively adjust the weights of various parameters, effectively avoiding limitations imposed by existing expert knowledge and experience, as well as subjective issues in weight determination. Data-driven health status assessment is seen as a balance between physical model-based assessment and expert experience-based assessment and is considered a major trend for future development [11]. In the realm of data-driven health status assessment methods, deep learning technology has achieved notable success in equipment health status recognition. In a study [12], a Bayesian network was combined with a large amount of monitoring data to construct a multi-objective

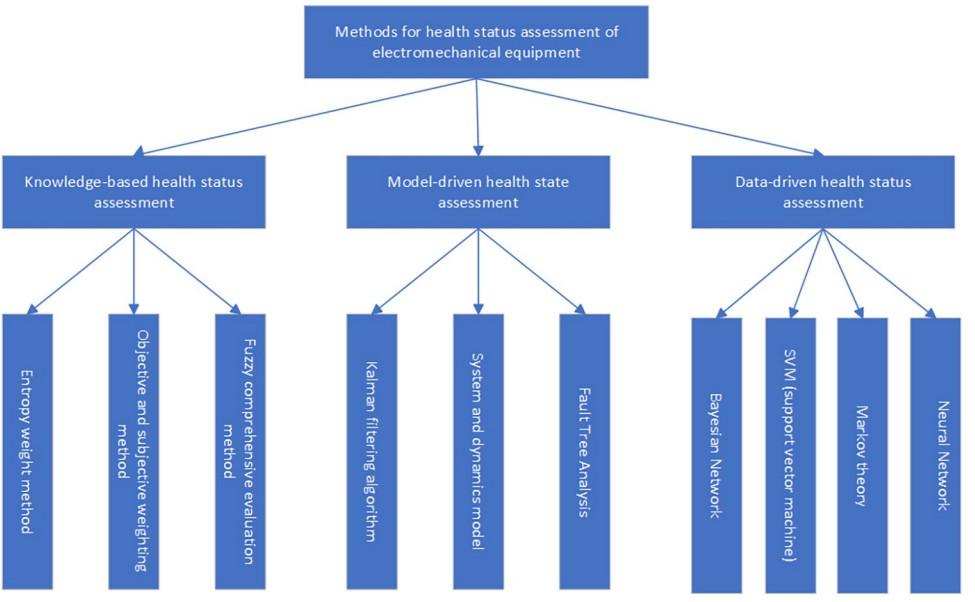

**Fig 1. Health status assessment methods.**

performance prediction model for a certain type of turbo-shaft engine, effectively predicting the engine's performance. Another study [13] used Long Short-Term Memory (LSTM) networks to identify the health status of wind turbines. The learning of network parameters employed the error backpropagation algorithm and Adagrad, achieving significant results in wind turbine health status recognition. An approach combining improved Ensemble Empirical Mode Decomposition (EEMD) and Hidden Markov Model (HMM) accurately diagnosed faults in the rocker arm bearings of a coal mining machine [14]. An enhanced regression-type Support Vector Machine (SVR) method accurately predicted the remaining life of rolling bearings [15]. A Bayesian Long Short-Term Memory Network (Bayesian-LSTM) was constructed for the health status recognition of external meshing gear pumps, achieving effective recognition [16].

To address the issues of susceptibility to expert knowledge and difficulty in establishing degradation models in the current health status assessment of scraper conveyors, this study utilizes historical and real-time operating data of scraper conveyors, employing deep learning techniques to leverage the fusion capabilities of multiple data sources and the efficient processing of multidimensional data. A one-dimensional convolutional neural network (1DCNN) is established as the health status perception model for scraper conveyors. Compared to commonly used models such as LSTM, although LSTM excels at capturing long-term dependencies, its higher number of parameters, longer training times, and susceptibility to overfitting limit its application in high real-time and complex industrial scenarios. This study chooses 1DCNN over LSTM and other models because 1DCNN can efficiently extract local features when processing multi-channel, high-frequency monitoring data, and has lower computational complexity, making it more suitable for real-time monitoring tasks [17–19]. Additionally, the application of Dropout layers further enhances the model's adaptability to noisy data, improves its generalization capability, and prevents overfitting, enabling the model to more accurately assess the health status of scraper conveyors and provide effective support and basis for maintenance management [20–22].

## Materials and methods

### Construction of a framework for health status perception of scraper conveyors

The construction of the framework for scraper conveyor health status perception is mainly divided into two major parts. The first part involves determining the health status monitoring parameters of the scraper conveyor and categorizing the corresponding health status levels for each monitoring parameter [23,24]. The health status of the scraper conveyor changes continuously with operating time, causing the monitored parameters of the equipment to undergo constant changes. Not all monitoring parameters may comprehensively reflect the current operational status of the equipment. Therefore, it is necessary to filter the monitoring parameters of the scraper conveyor and select those that can comprehensively reflect the current operational status of the equipment. Due to the multiple components of the scraper conveyor, a single monitoring parameter cannot reflect the health status information of the equipment. When using multiple monitoring parameters, advanced data fusion techniques are employed to accurately determine the health status of the scraper conveyor.

The second part involves the establishment of the health status perception model for the scraper conveyor. The model takes the monitoring parameters determined in the first step as input and outputs the categorized health status level of the scraper conveyor. The historical operational monitoring data of the scraper conveyor is then fed into the perception model with adjusted parameters. The constructed model autonomously learns the one-to-one

correspondence between various sets of monitoring data and health status levels. During the training process of the model, it calculates in real-time the gap between the current model output and the correct health status level. It continuously adjusts the distance between the model output health status level and the correct health status level based on the loss function value. This ensures that the final output of the model corresponds to the correct value. If the recognition accuracy of the model does not meet the specified requirements, indicating that the model cannot meet the needs of practical engineering, further adjustments are made to parameters such as the number of training iterations, learning rate, network layers in the constructed network model, the number of neurons in each hidden layer, and batch size during model training. This aims to improve the accuracy of the scraper conveyor health status perception model. Additionally, accurate information on recognition effectiveness can be obtained based on the accuracy provided after model training is completed.

## Selection and classification of health status parameters

The scraper conveyor is an indispensable long-distance coal transportation equipment underground, mainly composed of a transmission system and a conveyor section, as illustrated in Fig 2. The conveyor consists primarily of the head, tail, middle trough, transition section, scraper chain, shifting seat, and framework, among other components [25]. The transmission part of the equipment is mainly composed of a three-phase asynchronous electric motor, fluid coupling, reducer, blind shaft, scraper components, and other auxiliary components [26].

Through in-depth analysis of the composition, operation principles, historical fault information, and combining with the practical production conditions of the fully mechanized mining face and previous research results, it is evident that common faults in the scraper conveyor primarily occur in components such as the electric motor, reducer, and scraper chain [27–30]. To ensure that the selected monitoring parameters comprehensively characterize the health status of the scraper conveyor, the study combines the existing monitoring devices on the scraper conveyor. Parameters are chosen based on principles such as objectivity, measurability, completeness, independence, typicality, and consistency. Nine monitoring parameters have been selected, as shown in Table 1.

When dividing the health status levels of the scraper conveyor, the following characteristics need to be adhered to:

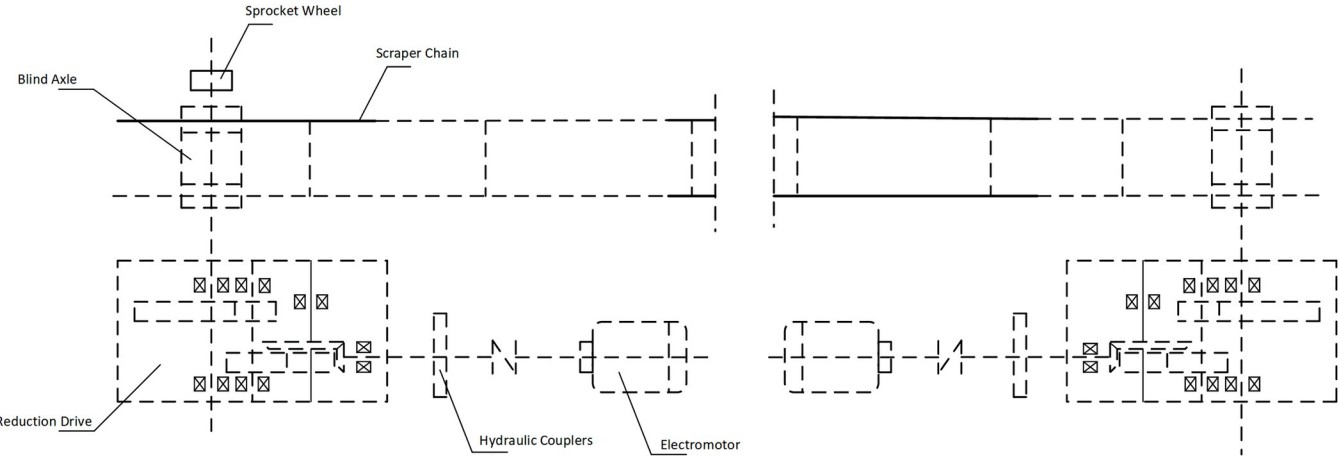

**Fig 2. Structural diagram of scraper conveyor.**

**Table 1. Scraper conveyor health monitoring parameters.**

| No. | Label | Component | Name |
|---|---|---|---|
| 1 | n1 | Electric Motor | Electric motor speed |
| 2 | T1 | Electric Motor | Electric motor temperature |
| 3 | I1 | Electric Motor | Electric motor current |
| 4 | V1 | Electric Motor | Electric motor vibration |
| 5 | T2 | Reducer | Reducer oil temperature |
| 6 | L1 | Reducer | Reducer oil level |
| 7 | Q1 | Cooling Water | Cooling water flow rate |
| 8 | P1 | Cooling Water | Cooling water pressure |
| 9 | P2 | Tensioning Cylinder | Tensioning cylinder pressure |

1. Specificity: The health level division is specific to the model of the scraper conveyor in this study. Simultaneously, the division should be tailored to the specific purpose, accurately determining the health status of the equipment.

2. Objectivity and Separability: In the process of level division, historical maintenance data and expert experience related to the equipment should be considered. It should follow the objective rules of equipment operation, with each health status corresponding to a reasonable range of equipment states. The division should incorporate various factors to reasonably classify the health status of the equipment.

3. Strategical Principles: For each health status determined for the equipment, specific maintenance strategies should be applicable. Following the principles of specificity, objectivity and separability, and strategical principles, the health status levels of the scraper conveyor have been determined, as shown in Table 2.

## Construction of a scraper conveyor health status perception model based on One-dimensional convolutional neural network(1DCNN)

**Convolutional neural network.** Convolutional Neural Network (CNN) is a special type of feedforward neural network that can be trained through backpropagation. Due to its distinctive structure, CNN has found significant applications in fields such as speech and image processing. The network extracts crucial features from the data through convolutional layers and compresses the retained information via pooling layers. As the data flows through the

**Table 2. Scraper conveyor health status level and description.**

| Equipment Health Status Level | Level Description | Level Label |
|---|---|---|
| Healthy | The equipment operates smoothly, with all components running normally and steadily. Only regular maintenance is needed. | 0 |
| Caution | The equipment's operational performance is generally stable, requiring regular scheduled maintenance. | 1 |
| Deterioration | The equipment's operational status is average, showing signs of deterioration. Immediate scheduling for maintenance is needed. | 2 |
| Fault | The equipment's monitoring indicators are abnormal, experiencing a malfunction, making it impossible to operate normally. Immediate shutdown for repair is required. | 3 |

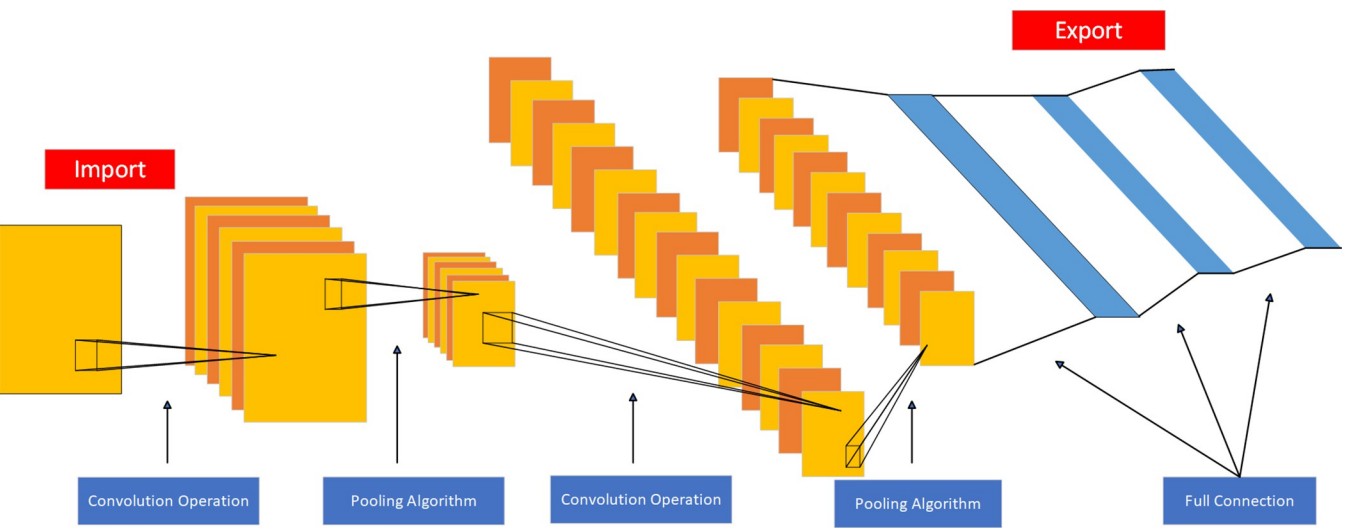

**Fig 3. Convolutional neural network structure diagram.**

network following its structure, undergoing multiple convolution and pooling operations, the model extracts topological features and acquires stable characteristics with rotation and translation invariance [31].

The traditional CNN consists of an input layer, convolutional layer, pooling layer, fully connected layer, and output layer, as illustrated in Fig 3. During training, CNN minimizes the loss function through gradient descent, adjusting network weights backward, and iteratively improving its accuracy. This process helps CNN optimize its classification performance for better predictive results. In this study, a one-dimensional convolutional neural network (1D-CNN) is used to construct the health status perception model for the scraper conveyor. 1D-CNN is commonly used for feature recognition and processing of single-dimensional time-series signals. It leverages the advantages of CNN, allowing the use of wider convolutional kernels for comprehensive feature extraction without increasing network parameters and computational complexity.

The convolutional layer consists of multiple convolutional kernels. Features embedded in the data are extracted through multiple convolution operations. After initializing the weights, the convolutional kernels move according to the stride. During movement, the weight parameters perform multiplication and addition operations with each digit corresponding to the sliding window. The output size can be adjusted by changing the stride and using zero-padding. The convolution operation is defined as follows:

$$c_i^{l+1}(j) = K_i^l * x^l(j) + b_i^l \tag{1}$$

where, $*$ represents the convolution between the kernel and the local region, $c_i^{l+1}(j)$ represents the input to the j-th neuron in the i-th channel of the layer $K_i^l$, $b_i^l$ represents the weights and bias of the i-th neuron for the l-th convolution kernel in the layer.

The pooling layer performs dimensionality reduction, with common types being max pooling and average pooling. Essentially, it is a down-sampling operation that transforms the input at a certain position in the network into the overall features of the data. In this study, max pooling is adopted, where the output at a certain position is the maximum value of the local

receptive field. Its mathematical description is as follows:

$$S_k^l(i) = \max_{(i-1)H+1 \leq t \leq iH} \{a_k^{l-1}(t)\} \qquad (2)$$

Where, $a_k^{l-1}(t)$ represents the activation value of the t-th neuron in the k-th feature vector of the l-1 layer; H represents the width of the pooling kernel; $S_k^l(i)$ represents the corresponding pooling result of the l layer.

The activation function connects the hidden layers in a neural network, preserving and mapping features of activated neurons in each layer. For some inseparable features in certain layers, it can map them to other layers, making them linearly separable. The use of activation functions enhances the expressive power of convolutional neural networks. In this study, ReLU is chosen as the activation function for the constructed neural network. ReLU, a typical piecewise function, determines the relationship between the input data and zero during computation. It converges quickly and better addresses issues such as gradient vanishing and exploding that may exist with other functions. The ReLU activation function is expressed as follows:

$$f(x) = \max(0, x) = \begin{cases} 0 & x < 0 \\ x & x \geq 0 \end{cases} \qquad (3)$$

The classification layer consists of a fully connected layer and a softmax classifier. The fully connected layer flattens the output of the pooling layer into a one-dimensional vector and processes concatenation. It is commonly used to extract correlations or feature representations between input data. The softmax function is a probability distribution strategy for multi-class problems. It maps the output of the model into a probability distribution, showing the predicted probability for each category and assisting in multi-label classification. Assuming a category label $y \in \{1, 2 \cdots, K\}$, the probability that a sample x belongs to class K is given by:

$$p(y = k \mid x; \Theta) = \text{softmax}(\theta_k^T x) = \frac{\exp(\theta_k^T x)}{\sum\limits_{j=1}^{K} \exp(\theta_k^T x)} \qquad (4)$$

where, $\Theta = [\theta_1, \theta_2, \cdots, \theta_K]$ represents all training parameters in the regression model, and $1/\sum\limits_{i=1}^{K} \exp(\theta_j^T x)$ is the normalization function.

## The 1D Convolutional neural network (1DCNN) health status model

**The structure of the 1d convolutional neural network(1dcnn) health status recognition model.** The convolutional neural network-based scraper conveyor health status recognition model designed in this study includes two convolutional layers and two pooling layers. The structure diagram is shown in Fig 4.

The first convolutional layer uses 16 convolutional kernels with a size of 71 to reduce the risk of overfitting and improve generalization ability and accuracy. The second convolutional layer uses 32 convolutional kernels with a size of 71. Both the first and second pooling layers use max pooling with a kernel size of 31 and a count of 7, with a stride of 13. A Dropout layer is added between the last pooling layer and the fully connected layer to prevent overfitting. The fully connected layer transforms the features obtained from the average pooling layer into a one-dimensional vector, which is then input into the subsequent Softmax layer for the classification of various health states of the scraper conveyor.

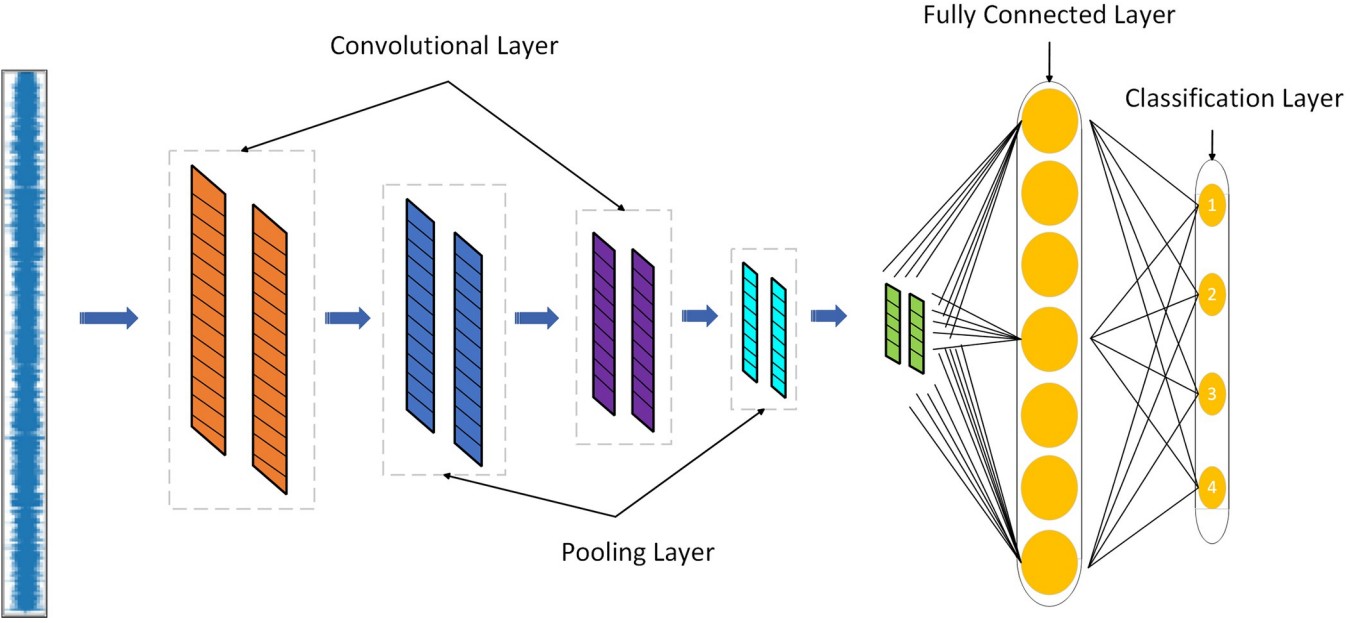

**Fig 4. 1DCNN health status identification model structure.**

**The workflow of 1d convolutional neural network (1dcnn) health status recognition.**
Firstly, collect monitoring data for various health status parameters of the scraper conveyor. Then, standardize the data to obtain input samples for the model. Subsequently, divide the samples into a training set and a testing set. The training set is utilized to train the model, enabling it to better fit the data and achieve higher accuracy. Meanwhile, the testing set is employed to assess the model's performance and evaluate its generalization ability.

The 1DCNN health status recognition process is divided into two stages: training and testing, as shown in Fig 5. After the dataset is divided, the model enters the training stage. In the training stage, the model's parameters are initialized. After parameter initialization, the model performs forward propagation to calculate the difference between the model's output and the true label values. The network parameters are then updated through backpropagation, reducing the error between them to meet the neural network's error requirements.

Finally, the model enters the validation stage to check if the accuracy on the testing set meets the requirements. If not, the model's hyperparameters are adjusted, and the training process continues until the desired accuracy on the testing set is achieved. The final result is the one-dimensional convolutional neural network model used for scraper conveyor health status assessment.

## Results and discussion

### Data acquisition and processing

This article focused on the scraper conveyor of a coal mining enterprise in Shanxi, China, selecting it as the research subject. A total of 10,023 real operational data points of the scraper conveyor under different health conditions were collected. These data include 3,300 instances of "healthy" states, 2,900 instances of "attention" states, 2,505 instances of "deterioration" states, and 1,318 instances of "fault" states. Each monitoring data point was standardized and corresponded one-to-one with its respective health state, forming input samples for the neural network.

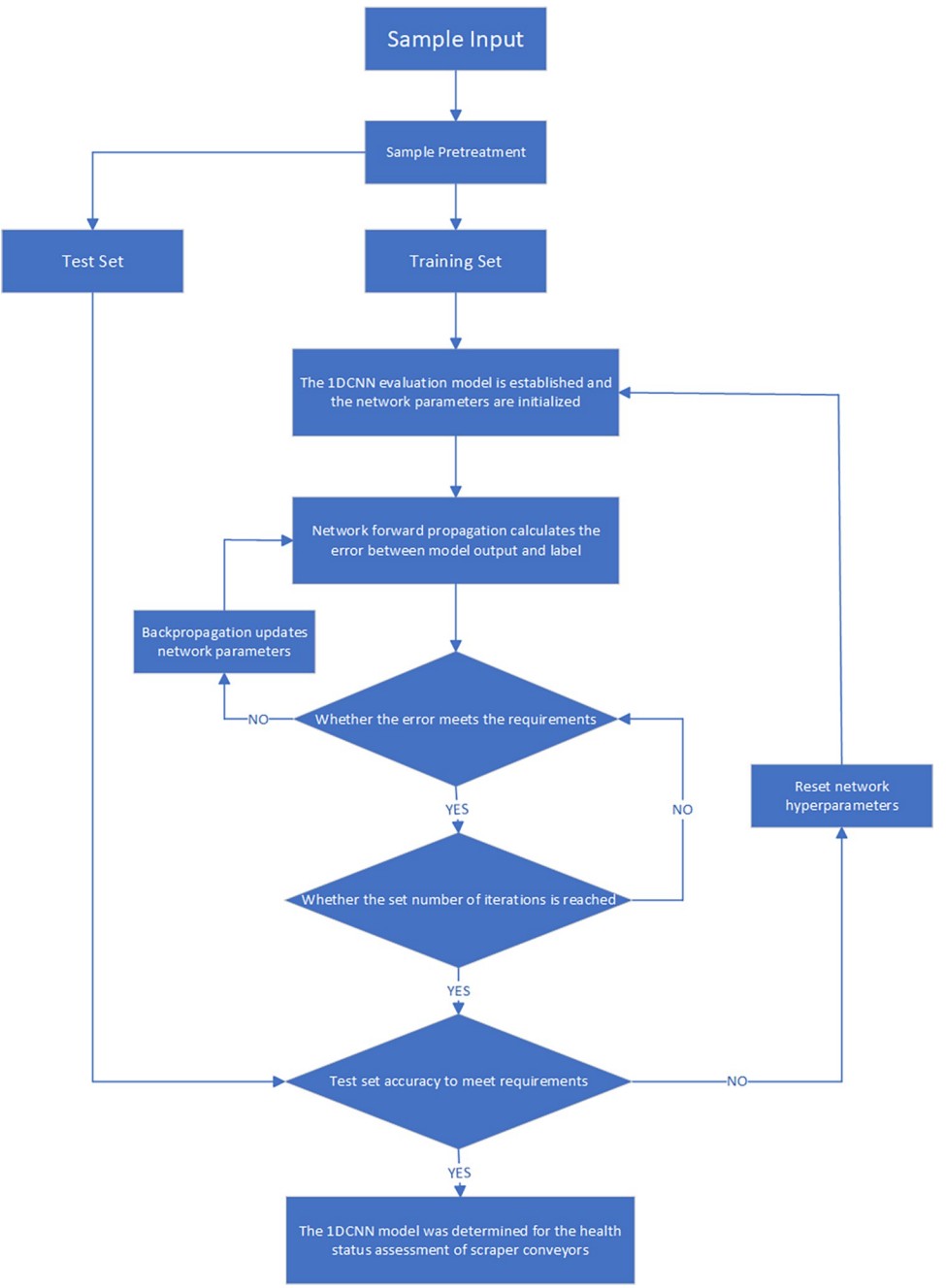

**Fig 5. 1DCNN health status identification process.**

To eliminate the influence of magnitudes and dimensions and enhance the model's accuracy, a mean normalization process was applied to the monitoring data. The mathematical representation of mean normalization is as follows:

$$x_{new} = \frac{x - \mu}{\sigma} \tag{5}$$

Where, μ is the normalized value, σ is the standard deviation of the sample data.

**Table 3. Standardized monitoring data.**

|  | n1 | T1 | I1 | V1 | T2 | L1 | Q1 | P1 | P2 |
|---|---|---|---|---|---|---|---|---|---|
| 1 | -1.80535 | 0.09587 | -1.40656 | -1.88692 | -1.89865 | 0.37241 | -0.27009 | -0.28684 | 1.57015 |
| 2 | -1.32001 | -1.01626 | 0.11389 | 0.34586 | -1.44128 | -0.01771 | -0.23872 | -0.28684 | -0.95646 |
| 3 | -2.16935 | 0.37390 | -2.31159 | -1.28650 | -0.94580 | -1.38316 | -0.39697 | 0.87807 | 1.57015 |
| 4 | -0.47068 | -0.34435 | -1.98578 | -0.36712 | -1.74619 | -1.64324 | -0.45257 | -0.86929 | 1.57015 |
| 5 | -0.92568 | -0.18216 | -2.38400 | -0.25455 | -1.44128 | -0.89550 | -0.23587 | 1.46052 | 1.25432 |
| . . . | . . . | . . . | . . . | . . . | . . . | . . . | . . . | . . . | . . . |
| 10020 | 0.68199 | -0.66872 | 0.25870 | -1.58671 | 0.50252 | -1.80580 | -0.14605 | 0.29561 | -0.64063 |
| 10021 | 0.56066 | -0.76140 | 0.04149 | -0.74238 | 0.88366 | 0.63249 | -0.11754 | -0.86929 | 0.93849 |
| 10022 | 0.65166 | -0.80774 | 0.15010 | 0.25204 | 0.73120 | -0.14775 | -0.17029 | -0.86929 | 1.57015 |
| 10023 | 0.56066 | -1.47966 | 0.00529 | -1.62423 | 0.88366 | -1.22061 | -0.14035 | 0.87807 | 0.30684 |
| 10024 | 0.59099 | 0.55926 | 0.04149 | -1.47413 | 0.84554 | -0.47286 | -0.06907 | 0.87807 | 1.254325 |

The data after standardized processing are shown in Table 3.

This study employed cross-validation to train and validate the health perception model of the scraper conveyor. The 10,023 real operating data of the scraper conveyor under different health conditions were split into a training set and a test set in an 8:2 ratio. The distribution of training and validation data for each health condition, along with their corresponding labels, is presented in Table 4.

## Model parameters and experimental validation

The 1DCNN model for recognizing the health status of the scraper conveyor, constructed in this study, includes two convolutional layers, two pooling layers, two Dropout layers, one fully connected layer, and one output layer. The model was built and trained using TensorFlow 2.9.1, Keras 2.9.0, and Python 3.9.12. After multiple training sessions, the optimal model parameters are determined as shown in Table 5.

After configuring the scraper conveyor health status recognition model based on Table 5, the learning rate of the model was adjusted to 0.001. The values of the training set and test set loss functions, as well as the accuracy of the model, are shown in Figs 6 and 7, respectively.

The figures indicate that the model stabilizes around 160 training iterations, with the training set accuracy plateauing at around 99.4% and the test set accuracy stabilizing at around 98.9%, meeting the requirements for scraper conveyor health status recognition.

To accurately observe the model's performance in each category, a confusion matrix of the model's predicted results was plotted. The confusion matrix of the model's predicted results is shown in Fig 8. The numbers 0 to 3 on the axes represent the four health states of the scraper conveyor. The confusion matrix provides further evaluation of the neural network's learning effectiveness. According to the confusion matrix, there are 22 mispredictions out of 2005 data points in the validation set, resulting in an average prediction accuracy of 98.9%. The accuracy of recognizing the "healthy" samples is 96%, "attention" samples is 99.6%, and both

**Table 4. Experimental sample of the health status perception model of scraper conveyor.**

| Sample Type | Training Set Samples | Test Set Samples | Label |
|---|---|---|---|
| Healthy | 1982 | 501 | 0 |
| Caution | 2015 | 521 | 1 |
| Deterioration | 2002 | 470 | 2 |
| Fault | 2019 | 513 | 3 |

**Table 5. Neural network parameters.**

| Sets | Values | Parameters | Values |
|---|---|---|---|
| Conv1D Kernel Size, Stride, and Number of Filters | 7×1、1×1、16 | Epochs | 200 |
| MaxPooling1D Pool Size and Stride | 2×1,2×1 | Batch Size | 128 |
| Dropout Rate | 0.1 | Activation Function | ReLu |
| Conv1D Kernel Size, Stride, and Number of Filters | 7×1、1×1、32 | Loss Function | Cross entropy loss |
| MaxPooling1D Pool Size and Stride | 2×1, 2×1 | Optimizer Type | Adam |
| Dropout Rate | 0.1 | Classifier | Softmax |

"deterioration" and "fault" samples have 100% accuracy. The lower accuracy in recognizing "healthy" samples may be attributed to the similarity between the "healthy" and "attention" health states, leading to misclassifications for certain samples by the model.

In order to further understand the learning capabilities of the 1DCNN scraper conveyor health status perception model built in this study for various health states, the T-SNE nonlinear dimensionality reduction algorithm [32] was selected. This algorithm was applied to reduce the original data of the test set as well as the data from the penultimate layer of the model to two dimensions for visualization and comparison. The visualized results after dimensionality reduction are shown in Figs 9 and 10.

Fig 9 shows the distribution of the original data samples. Due to the high correlation among the four health features in the original data, it is challenging to distinguish the four health state data, resulting in overlapping. Fig 10 represents the sample distribution after

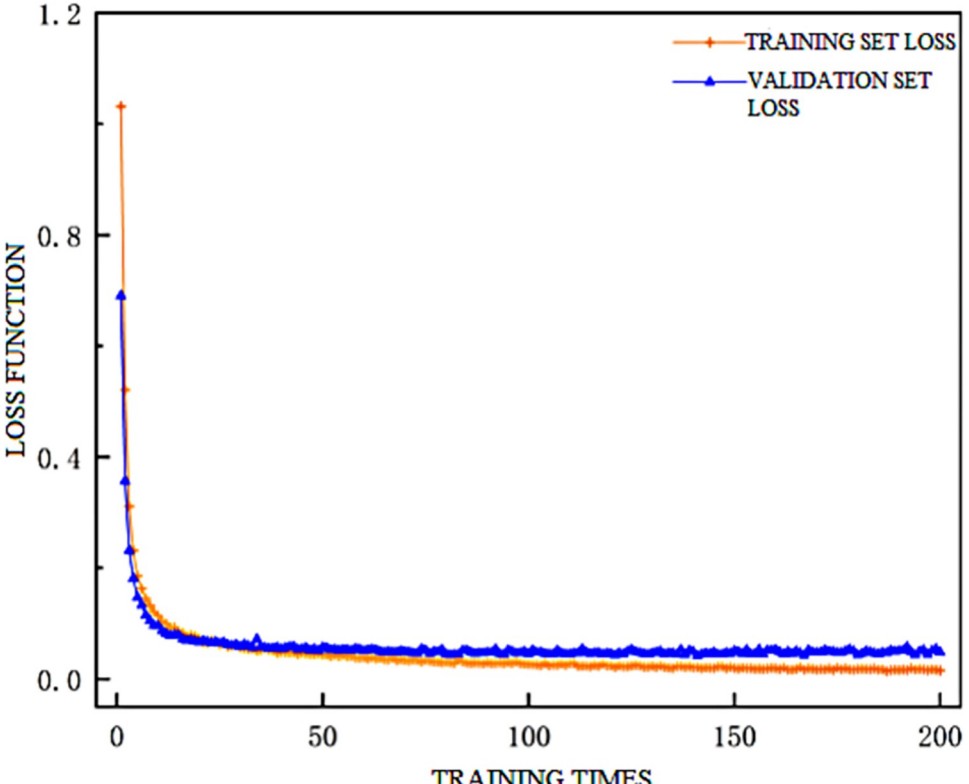

**Fig 6. Model training loss value graph.**

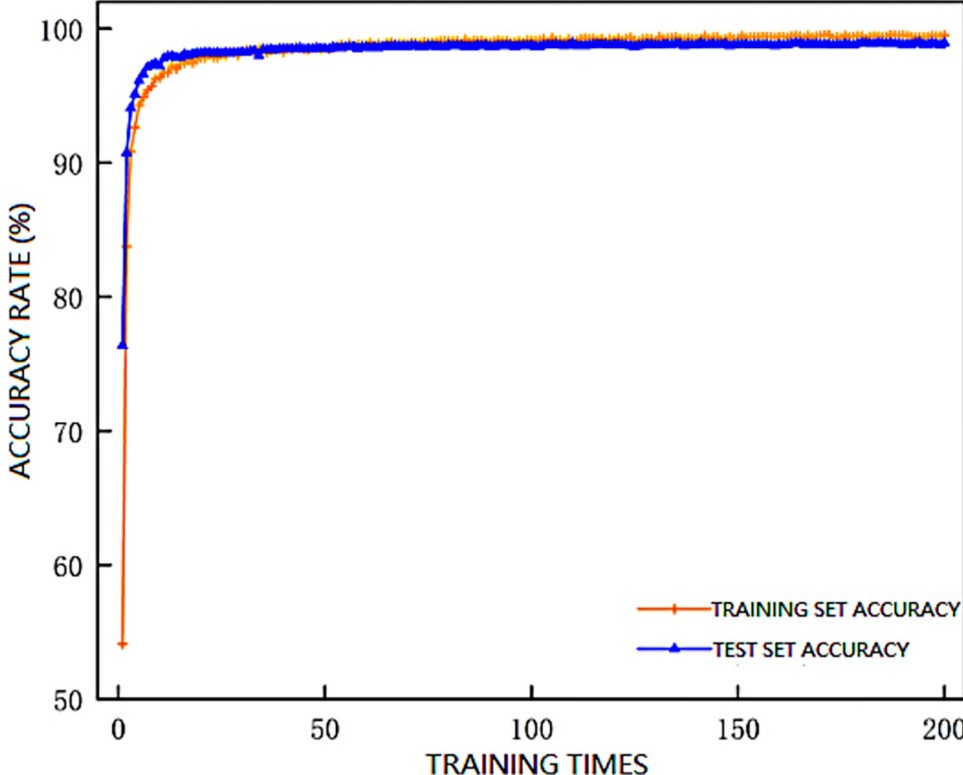

**Fig 7. Model training accuracy curve.**

feature extraction by 1DCNN. In this figure, samples of the four health states are generally concentrated in specific regions, demonstrating a more distinct separation among the four health states. Therefore, the 1DCNN scraper conveyor health status perception model proposed in this study is capable of effectively recognizing the health status of the scraper conveyor.

## Model comparison

The proposed 1D Convolutional Neural Network (1DCNN) was compared with other classification algorithms such as Ridge Classifier, Nearest Centroid, Perceptron, Decision Tree Classifier, and Multi-Layer Perceptron (MLP). Through repeated experiments and calculating the average accuracy of each model, the results indicate that the 1DCNN model proposed in this study has higher accuracy and is more suitable as a health perception model for scraper conveyors. The accuracy of each algorithm model on the training and test sets is shown in Table 6.

## Conclusions

For the current challenges in scraper conveyor health assessment, such as susceptibility to expert knowledge and the difficulty in establishing equipment degradation models, a method for scraper conveyor health state perception based on one-dimensional convolutional neural networks (1DCNN) was proposed. The feasibility of this method was verified by recognizing the health states using real operational data from a coal mining enterprise in Shanxi. The conclusions drawn from this study are as follows:

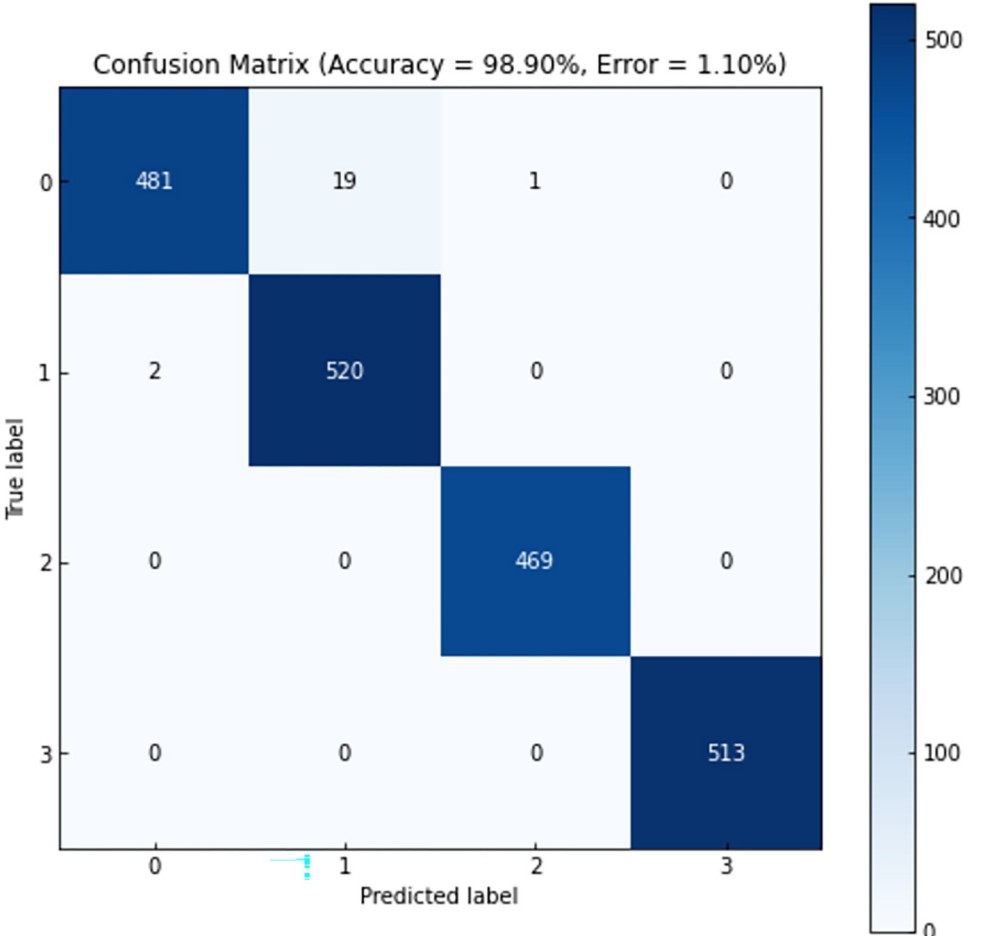

**Fig 8. Health status identification effect confusion matrix.**

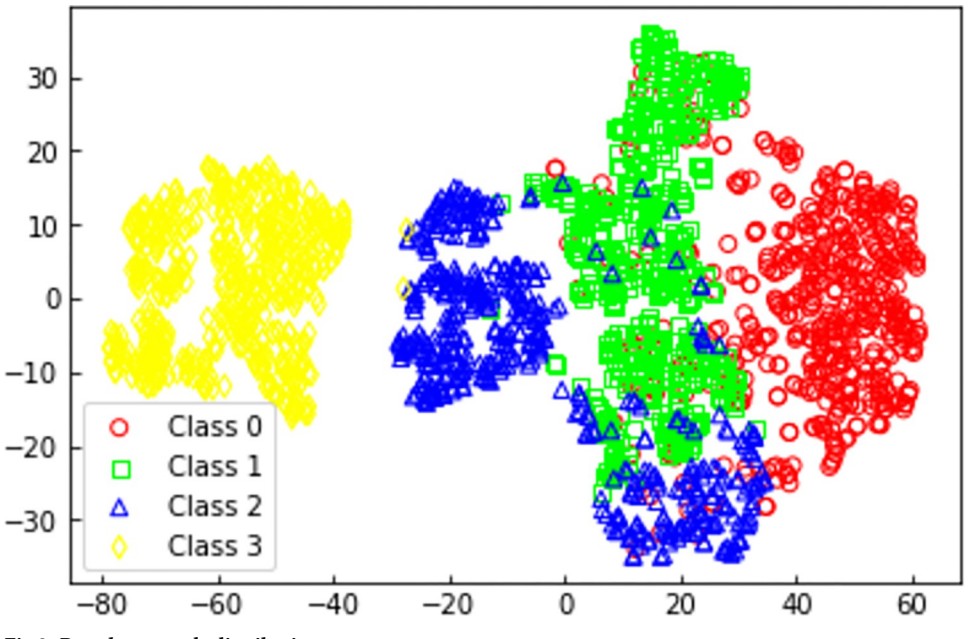

**Fig 9. Raw data sample distribution.**

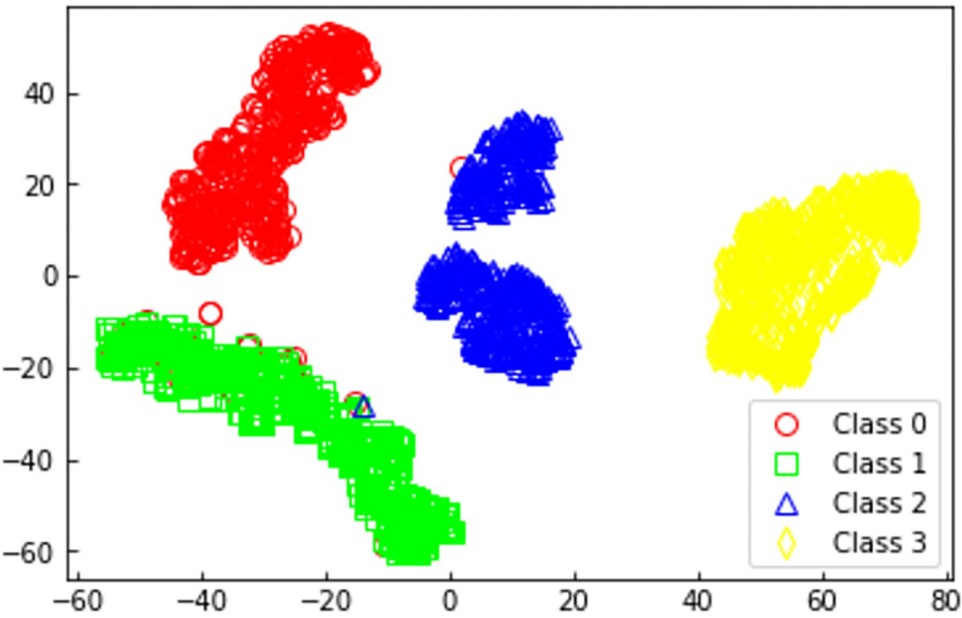

**Fig 10. 1DCNN sample distribution after feature extraction.**

**Table 6. The accuracy rate of each algorithm model training set and test set.**

| Model Name | Average Accuracy (%) | |
|---|---|---|
| | **Training Set** | **Test Set** |
| Ridge Classifier | 87.43 | 86.93 |
| Nearest Centroid | 89.32 | 88.87 |
| Perceptron | 91.23 | 90.97 |
| DecisionTree Classifier | 96.51 | 95.36 |
| Multi-Layer Perception, MLP | 97.13 | 96.41 |
| 1DCNN | 99.42 | 98.93 |

The proposed scraper conveyor health state perception model can adaptively learn effective features for equipment health state recognition from raw monitoring data, avoiding the influence of expert experience on weight determination and achieving high accuracy.

The scraper conveyor health state perception method based on 1DCNN effectively combines feature extraction and health state classification. Additionally, the model achieves short recognition times after training, enabling real-time health state recognition for scraper conveyors end-to-end, demonstrating practicality in scraper conveyor health state recognition.

While the method shows some effectiveness in recognizing scraper conveyor real data with noise, the harsh environment in the comprehensive mining face introduces strong noise in the monitored equipment status data. Future research will focus on studying denoising techniques for strong noise in monitoring data to enhance the robustness of this method.

## Supporting information

**S1 Data set.**
(XLS)

**S1 Graphical abstract.**
(TIF)

## Author Contributions

**Data curation:** Zhenlin Liu, Chenhui Han, Zhiqiang Yang, Jialu Zheng.

**Investigation:** Zhenlin Liu, Wangjie Zhang.

**Methodology:** Jie Lu.

**Software:** Jialu Zheng.

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
