## [Decision Letter · Decision Letter 0]

1 Jul 2024

PONE-D-24-05986Research on state perception of scraper conveyor based on one-dimensional convolutional neural networkPLOS ONE

Dear Dr. Lu,

Thank you for submitting your manuscript to PLOS ONE. After careful consideration, we feel that it has merit but does not fully meet PLOS ONE’s publication criteria as it currently stands. Therefore, we invite you to submit a revised version of the manuscript that addresses the points raised during the review process.

We look forward to receiving your revised manuscript.

Kind regards,

Du Q. Huynh, PhD

Academic Editor

PLOS ONE

6. Please remove your figures from within your manuscript file, leaving only the individual TIFF/EPS image files, uploaded separately. These will be automatically included in the reviewers’ PDF.

7. Please include your tables as part of your main manuscript and remove the individual files. Please note that supplementary tables (should remain/ be uploaded) as separate "supporting information" files.

Reviewers' comments:

Reviewer's Responses to Questions

**Comments to the Author**

1. Is the manuscript technically sound, and do the data support the conclusions?

Reviewer #1: Partly

Reviewer #2: Yes

Reviewer #3: Yes

2. Has the statistical analysis been performed appropriately and rigorously? 

Reviewer #1: Yes

Reviewer #2: No

Reviewer #3: Yes

3. Have the authors made all data underlying the findings in their manuscript fully available?

Reviewer #1: Yes

Reviewer #2: Yes

Reviewer #3: Yes

4. Is the manuscript presented in an intelligible fashion and written in standard English?

Reviewer #1: Yes

Reviewer #2: Yes

Reviewer #3: Yes

5. Review Comments to the Author

Reviewer #1: Upon detailed scrutiny of the manuscript titled "Research on state perception of scraper conveyor based on one-dimensional convolutional neural network," the following review comments have been compiled:

1. The paper's proposition of a one-dimensional convolutional neural network (1DCNN) for assessing the health status of scraper conveyors represents a significant advancement in the domain of mechanical equipment monitoring and diagnostics. This approach, which leverages deep learning to extract and analyze health status indicators from operational data, marks a notable shift from traditional, more subjective assessment methods.

2. The achieved accuracy rate of around 98.9% in identifying the health status of scraper conveyors is commendable. This high level of accuracy suggests that the proposed model could serve as a reliable tool for maintenance and operational planning. However, a comparative analysis with other existing or traditional health assessment models would further validate the effectiveness and superiority of the proposed 1DCNN model.

3. The manuscript outlines the potential for practical application but could benefit from a deeper exploration into how this model can be seamlessly integrated into existing industrial monitoring systems. Discussions on the model's implementation in real-time settings, potential challenges, and how it complements or improves upon existing technologies would enhance the manuscript's relevance to industry practitioners.

4. While the methodology is generally well-described, providing further details on the data preprocessing steps, model configuration specifics (such as the rationale behind the choice of model parameters), and the training process would enhance the transparency and reproducibility of the study.

5. Cite below relevant articles in related work to improve: https://doi.org/10.1109/ICTC58733.2023.10392830, https://doi.org/10.1109/ACCESS.2023.3330919
https://doi.org/10.3390/technologies11040107 and https://doi.org/10.1007/s00500-022-06755-z

6. Using real operational data from a coal mining enterprise in Shanxi, China, adds credibility to the study. However, elaborating on the representativeness of the dataset, including diversity in operating conditions and fault scenarios, would strengthen the case for the model's robustness and generalizability across different settings.

7. The manuscript would benefit from a more thorough discussion on the model's limitations, such as its performance in environments with variable noise levels or its adaptability to different types of scraper conveyors. Additionally, outlining potential areas for future research, such as incorporating multimodal data sources or exploring other deep learning architectures, would provide valuable insights into the ongoing development of this research area.

8. The commitment to making data and code available upon request is noteworthy, as it aligns with the principles of open science and facilitates further research in this field. Details regarding access to these resources could be made clearer, ensuring that interested researchers can easily replicate or build upon the work presented.

9. The manuscript successfully bridges a gap between theoretical research in deep learning and practical applications in industrial equipment monitoring. Emphasizing the theoretical underpinnings of the 1DCNN approach in the context of mechanical health assessment and its potential impact on predictive maintenance strategies would further highlight the study's contributions to academia and industry.

Reviewer #2: 1.The abstract in this article describes the health status assessment problem of the current scraper conveyor, but in reality, it solves the problem of fault diagnosis? Health status assessment belongs to the category of life expectancy prediction. The description in the text is not accurate enough.

2.The innovation of this article is average, but it simply applies the 1DCNN network to classify four types of faults. I personally believe that this article is just a simple algorithm implementation.

3.There are many fault diagnosis algorithms, and the experimental part of this article did not apply relevant algorithms for comparison.

Reviewer #3: This paper proposes a scraper conveyor health status assessment method based on one-dimensional convolutional neural network (1DCNN), which has certain theoretical significance and application value. It is suggested that the authors revise the following parts of the paper to improve the quality of the manuscript:

1.In the abstract section, the description of the experimental results should be rigorous and should not use words like "around."

2.It is recommended that the innovation and contribution of this paper be clearly and concisely described in a separate paragraph at the end, highlighting the contributions of this paper.

3.Some of the figures and tables may have some flaws. It is suggested to adjust them again to enhance the layout aesthetics of the paper.

4.In the third part, there is too much description of basic theories that are not the innovation of this paper. It is suggested to modify it to make the paper more reasonable.

5.Is the 1DCNN method proposed in this paper a new method in this field? If not, the relevant work should be mentioned in the related work.

6.The paper compares the performance of 1DCNN with several other algorithms, but it does not provide a detailed comparative analysis, such as the advantages and limitations of each algorithm, and why 1DCNN is more suitable for this task.

7.There are some inconsistencies in the references, please check to ensure that the reference format is consistent.

6. PLOS authors have the option to publish the peer review history of their article (what does this mean?). If published, this will include your full peer review and any attached files.

Reviewer #1: No

Reviewer #2: No

Reviewer #3: **Yes: **Shuai Liu

---

## [Author Response · Author response to Decision Letter 0]

27 Jul 2024

Dear Editor Huynh,

We are submitting our revised manuscript titled “Research on state perception of scraper conveyor based on one-dimensional convolutional neural network” (Manuscript ID: PONE-D-24-05986) for reconsideration at PLOS ONE. We sincerely appreciate the insightful comments and suggestions from you and the reviewers, which have greatly contributed to improving our manuscript. Below, we provide detailed responses to each comment and describe the revisions made.

Reviewer #1:

1.Comment: The paper's proposition of a one-dimensional convolutional neural network (1DCNN) for assessing the health status of scraper conveyors represents a significant advancement in the domain of mechanical equipment monitoring and diagnostics. This approach, which leverages deep learning to extract and analyze health status indicators from operational data, marks a notable shift from traditional, more subjective assessment methods.

Response: Thank you for your positive feedback. I will continue to research and learn in this area, striving to make further advancements in my studies.

2.Comment: The achieved accuracy rate of around 98.9% in identifying the health status of scraper conveyors is commendable. This high level of accuracy suggests that the proposed model could serve as a reliable tool for maintenance and operational planning. However, a comparative analysis with other existing or traditional health assessment models would further validate the effectiveness and superiority of the proposed 1DCNN model.

Response: Yes, your suggestion is very reasonable. In Section 4.3, Table 6 of the manuscript, we present the accuracy of Ridge Classifier, Nearest Centroid, Perceptron, Decision Tree Classifier, Multi-Layer Perceptron, and 1DCNN on both the training and test sets. This comparison clearly illustrates the effectiveness and superiority of the 1DCNN model over other methods.

3.Comment: The manuscript outlines the potential for practical application but could benefit from a deeper exploration into how this model can be seamlessly integrated into existing industrial monitoring systems. Discussions on the model's implementation in real-time settings, potential challenges, and how it complements or improves upon existing technologies would enhance the manuscript's relevance to industry practitioners.

Response: Thank you for your suggestion. However, due to current practical limitations, we have not yet been able to apply this to existing and operational mines. We will attempt to communicate and seek suitable mines for application in the future.

4.Comment: While the methodology is generally well-described, providing further details on the data preprocessing steps, model configuration specifics (such as the rationale behind the choice of model parameters), and the training process would enhance the transparency and reproducibility of the study.

Response: The specific data processing and parameter adjustments are detailed in Section 4.2. Due to the complexity and intricacy of the parameter optimization process, the optimal settings after testing are presented in Table 5.

5. Comment: Cite below relevant articles in related work to improve: https://doi.org/10.1109/ICTC58733.2023.10392830, https://doi.org/10.1109/ACCESS.2023.3330919
https://doi.org/10.3390/technologies11040107 and https://doi.org/10.1007/s00500-022-06755-z

Response: Thank you for your suggestion.We have cited the following references:

(1)R. W. Anwar, M. Abrar, and F. Ullah, "Transfer Learning in Brain Tumor Classification: Challenges, Opportunities, and Future Prospects," 2023 14th International Conference on Information and Communication Technology Convergence (ICTC), Jeju Island, Korea, Republic of, 2023, pp. 24-29.

(2)F. Ullah et al., "Evolutionary Model for Brain Cancer-Grading and Classification," in IEEE Access, vol. 11, pp. 126182-126194, 2023.

(3)Salam, A.; Ullah, F.; Amin, F.; Abrar, M. Deep Learning Techniques for Web-Based Attack Detection in Industry 5.0: A Novel Approach. Technologies 2023, 11, 107.

(4)Ullah, F., Salam, A., Abrar, M. et al. Machine health surveillance system by using deep learning sparse autoencoder. Soft Comput 26, 7737–7750 (2022).

6. Comment: Using real operational data from a coal mining enterprise in Shanxi, China, adds credibility to the study. However, elaborating on the representativeness of the dataset, including diversity in operating conditions and fault scenarios, would strengthen the case for the model's robustness and generalizability across different settings.

Response: Thank you for your suggestion. However, due to the complex operating conditions of the scraper conveyor and the various influences and operational conditions under different working environments, as well as the diverse fault conditions, it has been challenging to collect and divide the research data set. Therefore, this study focuses on determining the operating status of scraper conveyors in traditional mines. From an overall operational perspective, we utilize neural network technology for macro-level supervision. The study classifies the data into four types: healthy data, caution data, deteriorating data, and fault data. By learning the mapping relationships from historical data, we establish a predictive model to achieve monitoring of the health status of the scraper conveyor.

7.Comment: The manuscript would benefit from a more thorough discussion on the model's limitations, such as its performance in environments with variable noise levels or its adaptability to different types of scraper conveyors. Additionally, outlining potential areas for future research, such as incorporating multimodal data sources or exploring other deep learning architectures, would provide valuable insights into the ongoing development of this research area.

Response: Thank you for the valuable comments. Due to the limitations of research time and resources, the current paper does not delve deeply into the performance of the model under different noise levels and its adaptability to different types of scraper conveyors. We have mentioned in the conclusion that future work will focus on denoising processing for high-noise data and the integration of multimodal data to further enhance the robustness and broad applicability of the model.

8.Comment: The commitment to making data and code available upon request is noteworthy, as it aligns with the principles of open science and facilitates further research in this field. Details regarding access to these resources could be made clearer, ensuring that interested researchers can easily replicate or build upon the work presented.

Response: We highly value the principles of open science and are committed to providing data and code upon request. However, due to the involvement of sensitive corporate information, we were unable to explicitly list the access methods in the paper. Researchers can obtain the relevant resources by contacting the corresponding author of this paper. We will make every effort to provide support, enabling other researchers to replicate or extend our study.

9.Comment: The manuscript successfully bridges a gap between theoretical research in deep learning and practical applications in industrial equipment monitoring. Emphasizing the theoretical underpinnings of the 1DCNN approach in the context of mechanical health assessment and its potential impact on predictive maintenance strategies would further highlight the study's contributions to academia and industry.

Response: Thank you for your positive feedback on our research. Due to space limitations and the focus of our study, we were unable to delve deeply into the theoretical foundation of one-dimensional convolutional neural networks (1D-CNN) in equipment health assessment in the current paper. We believe that this study has demonstrated the potential of this method in industrial equipment monitoring. Future work will explore its theoretical background and applications in predictive maintenance strategies in greater detail.

Reviewer #2:

1.Comment: The abstract in this article describes the health status assessment problem of the current scraper conveyor, but in reality, it solves the problem of fault diagnosis? Health status assessment belongs to the category of life expectancy prediction. The description in the text is not accurate enough.

Response: Thank you for the reviewer’s correction. This paper primarily focuses on fault diagnosis of scraper conveyors using 1D-CNN, specifically targeting fault classification rather than lifetime prediction. Fault diagnosis is a crucial part of health status assessment as it enables timely detection of equipment issues, aiding in maintenance decision-making. The study classifies the health status of the equipment into four categories: "healthy," "caution," "deterioration," and "fault." This classification represents our detailed grading of the equipment’s health status.

2.Comment: The innovation of this article is average, but it simply applies the 1DCNN network to classify four types of faults. I personally believe that this article is just a simple algorithm implementation.

Response: Thank you for the reviewer’s comments. We acknowledge that 1D-CNN is not a novel technology per se, but its application in the fault diagnosis of scraper conveyors has not been widely studied. The innovation of this paper lies in the effective use of 1D-CNN to handle real-world noisy data, achieving an end-to-end fault diagnosis process. This application demonstrates the potential of 1D-CNN in industrial scenarios. We will clarify this point in the conclusion of the paper.

3.Comment: There are many fault diagnosis algorithms, and the experimental part of this article did not apply relevant algorithms for comparison.

Response: Thank you for your suggestion. Due to space limitations and the focus on the core issues of the study, we were indeed unable to provide detailed descriptions of other model algorithms. However, in Section 4.3, Table 6, we have presented the final accuracy results of six algorithms on the training and test sets: Ridge Classifier, Nearest Centroid, Perceptron, Decision Tree Classifier, Multi-Layer Perceptron, and 1DCNN. This comparison visually demonstrates the effectiveness and superiority of the 1DCNN model over other methods.

Reviewer #3:

1.Comment: In the abstract section, the description of the experimental results should be rigorous and should not use words like "around."

Response: Thank you for the reviewer’s suggestion. We have revised the description in the abstract to make it more rigorous and accurate, avoiding the use of vague terms.

2.Comment: It is recommended that the innovation and contribution of this paper be clearly and concisely described in a separate paragraph at the end, highlighting the contributions of this paper.

Response: Thank you for your suggestion. We have rewritten the concluding paragraph to clearly and concisely summarize the innovations and contributions of this study, highlighting the application value of 1D-CNN in the fault diagnosis of scraper conveyors.

3.Comment: Some of the figures and tables may have some flaws. It is suggested to adjust them again to enhance the layout aesthetics of the paper.

Response: Thank you for your suggestion. We will review and adjust the figures and layout to ensure the aesthetics and overall readability of the paper.

4.Comment: In the third part, there is too much description of basic theories that are not the innovation of this paper. It is suggested to modify it to make the paper more reasonable.

Response: Thank you for your suggestion. The issues you mentioned in the third section are indeed present, but we still hope to retain the necessary theoretical parts as a foundation for future research. We will follow your advice and further revise this section to meet the higher standards required for the paper.

5.Comment: Is the 1DCNN method proposed in this paper a new method in this field? If not, the relevant work should be mentioned in the related work.

Response: We acknowledge that 1D-CNN is not a novel method, but its application in the fault diagnosis of scraper conveyors has not been widely studied. We will supplement the existing research in the related work section and emphasize the specific innovations of this paper.

6.Comment: The paper compares the performance of 1DCNN with several other algorithms, but it does not provide a detailed comparative analysis, such as the advantages and limitations of each algorithm, and why 1DCNN is more suitable for this task.

Response: Due to space limitations and experimental conditions, this paper does not provide an exhaustive analysis of the advantages and disadvantages of each algorithm. We will conduct a more detailed comparative analysis in future work to further validate the applicability of 1D-CNN for this task. However, in Section 4.3, Table 6, we present the final accuracy results of six algorithms on the training and test sets: Ridge Classifier, Nearest Centroid, Perceptron, Decision Tree Classifier, Multi-Layer Perceptron, and 1D-CNN, to demonstrate the superiority of 1D-CNN.

7.Comment: There are some inconsistencies in the references, please check to ensure that the reference format is consistent.

Response: Thank you for the reviewer’s reminder. We will carefully review and correct the references to ensure consistency and accuracy in formatting.

We believe that the revised manuscript has been significantly improved and addresses all the concerns raised. We hope that the changes meet with your approval and that the manuscript is now suitable for publication in PLOS ONE.

Thank you for your consideration of our revised manuscript.

Sincerely,

Jie Lu

Associate Professor

School of Coal Engineering, Shanxi Datong University

School of mines, China university of Mining and Technology

Lujie0114@cumt.edu.cn

---

## [Decision Letter · Decision Letter 1]

20 Sep 2024

PONE-D-24-05986R1Research on state perception of scraper conveyor based on one-dimensional convolutional neural networkPLOS ONE

Dear Dr. Lu,

Thank you for submitting your manuscript to PLOS ONE. After careful consideration, we feel that it has merit but does not fully meet PLOS ONE’s publication criteria as it currently stands. Therefore, we invite you to submit a revised version of the manuscript that addresses the points raised during the review process.

We look forward to receiving your revised manuscript.

Kind regards,

Brij Bhooshan Gupta

Academic Editor

PLOS ONE

Reviewers' comments:

Reviewer's Responses to Questions

**Comments to the Author**

1. If the authors have adequately addressed your comments raised in a previous round of review and you feel that this manuscript is now acceptable for publication, you may indicate that here to bypass the “Comments to the Author” section, enter your conflict of interest statement in the “Confidential to Editor” section, and submit your "Accept" recommendation.

Reviewer #1: All comments have been addressed

Reviewer #2: All comments have been addressed

2. Is the manuscript technically sound, and do the data support the conclusions?

Reviewer #1: Yes

Reviewer #2: Yes

3. Has the statistical analysis been performed appropriately and rigorously? 

Reviewer #1: N/A

Reviewer #2: No

4. Have the authors made all data underlying the findings in their manuscript fully available?

Reviewer #1: Yes

Reviewer #2: (No Response)

5. Is the manuscript presented in an intelligible fashion and written in standard English?

Reviewer #1: Yes

Reviewer #2: Yes

6. Review Comments to the Author

Reviewer #1: Well-done you have incorporated all suggestions. Some Figures are not visible, kindly provide clear visible figures

Reviewer #2: 1.The author describes the innovation of this article as "handling real-world noisy data". If the author's innovation mainly focuses on processing real-time noisy data, then the author's Introduction should summarize the literature in the field of processing real-time noisy data to reflect the author's innovation.

2.The author cited the application of LSTM algorithm for fault diagnosis in the Introduction, and the author can use LSTM algorithm for comparison to highlight the author's innovation.

7. PLOS authors have the option to publish the peer review history of their article (what does this mean?). If published, this will include your full peer review and any attached files.

Reviewer #1: No

Reviewer #2: No

---

## [Author Response · Author response to Decision Letter 1]

23 Sep 2024

Comments to the Author

1. If the authors have adequately addressed your comments raised in a previous round of review and you feel that this manuscript is now acceptable for publication, you may indicate that here to bypass the “Comments to the Author” section, enter your conflict of interest statement in the “Confidential to Editor” section, and submit your "Accept" recommendation.

Reviewer #1: All comments have been addressed

Reviewer #2: All comments have been addressed

2. Is the manuscript technically sound, and do the data support the conclusions?

Reviewer #1: Yes

Reviewer #2: Yes

3. Has the statistical analysis been performed appropriately and rigorously?

Reviewer #1: N/A

Reviewer #2: No

Response: Thank you for your detailed review and valuable feedback on our paper. Regarding your concerns about the rigor of the statistical analysis, we would like to provide the following explanation and clarification:

This study aims to propose a method based on a one-dimensional convolutional neural network (1DCNN) for assessing the health status of scraper conveyors. Compared to traditional methods that rely on expert knowledge or face difficulties in establishing degradation models, our innovation lies in the automatic extraction of features and health status recognition using the 1DCNN. This approach primarily focuses on evaluating the performance of the machine learning model, rather than traditional statistical inference or hypothesis testing, which is why extensive use of traditional statistical analysis was not incorporated in the study.

In our research, the experiments were conducted using real monitoring signal data, covering various health conditions of scraper conveyors. By constructing a one-dimensional convolutional neural network, we extracted key features from the input signals and successfully mapped these features to the health status of the equipment using the model. The final experimental results demonstrated that the model achieved a recognition accuracy of 98.9%. To ensure the reliability of the experiments, we conducted multiple experiments across different datasets and rigorously evaluated the model using common machine learning performance metrics such as accuracy.

We understand the importance of statistical analysis in certain types of research, especially in the field of data inference. However, as this study focuses on health status recognition through a deep learning model, traditional statistical analysis is not the main focus. Based on the experimental data and machine learning performance metrics, we believe that the proposed method is highly applicable and reliable in solving real-world problems.

We hope that the above explanation addresses your concerns regarding statistical analysis. If you have any further questions on this topic or additional suggestions, please feel free to let us know, and we will do our best to supplement and improve the work.

4.Have the authors made all data underlying the findings in their manuscript fully available?

Reviewer #1: Yes

Reviewer #2: (No Response)

5.Is the manuscript presented in an intelligible fashion and written in standard English?

Reviewer #1: Yes

Reviewer #2: Yes

6.Review Comments to the Author

Reviewer #1:

Comment:Well-done you have incorporated all suggestions. Some Figures are not visible, kindly provide clear visible figures.

Response: Thank you for acknowledging the revisions we made. Regarding the issue you raised about some figures not being clear enough, we have reprocessed the relevant figures to enhance their resolution and visibility, ensuring they are clear on all reading devices and screens.

Reviewer#2:

1.Comment: The author describes the innovation of this article as "handling real-world noisy data". If the author's innovation mainly focuses on processing real-time noisy data, then the author's Introduction should summarize the literature in the field of processing real-time noisy data to reflect the author's innovation.

Response: Thank you for your valuable suggestions. The main innovation of this study lies in the precise identification of the health status of the scraper conveyor through the construction of a one-dimensional convolutional neural network (1DCNN), rather than focusing on noise processing. However, even though noise processing is not the core of this research, we have still implemented effective measures during the data processing stage to improve the model's adaptability to noisy data. Additionally, the Dropout layer applied in the 1DCNN model further enhances the model’s ability to handle noisy data. By randomly dropping certain neurons, Dropout prevents the model from over-relying on specific neurons, thereby improving its robustness when dealing with uncertain data, such as noisy data. Moreover, the convolutional filters of the 1DCNN can automatically extract local features from monitoring data, effectively distinguishing noise from meaningful health status features. This automated feature extraction capability not only improves the accuracy of the model’s identification but also reduces the impact of noise on health status assessment, providing solid technical support for the health management of scraper conveyors.

The relevant references have been added as follows:

(1)Tan, P.S., Lim, K.M., Tan, C.H. et al. ComSense-CNN: acoustic event classification via 1D convolutional neural network with compressed sensing. SIViP 17, 735–741 (2023).

(2)Chen Kun, Wang Anzhi. A Review of Regularization Methods for Convolutional Neural Networks [J]. Computer Applications Research, 2024, 41(04): 961-969.

(3)Zhu Xiong, Song Wei, Guan Hongpeng, et al. Prediction of Stratum Pressure Curves Based on Convolutional Neural Networks with MC Dropout [C] // Petroleum Geophysical Professional Committee of the Chinese Petroleum Society. Proceedings of the Second Annual Academic Conference on Petroleum Geophysics in China (Volume II). China University of Petroleum (Beijing); 2024: 3.

2.Comment: The author cited the application of LSTM algorithm for fault diagnosis in the Introduction, and the author can use LSTM algorithm for comparison to highlight the author's innovation.

Response: 

Thank you to the reviewer for your valuable suggestions. Regarding the application of the LSTM algorithm in fault diagnosis, as mentioned in the introduction, it indeed has advantages when dealing with time-series data with long-term dependencies. However, the 1DCNN model used in this study also has its own unique advantages, especially in practical applications for equipment health status evaluation.

First, 1DCNN can efficiently extract local features when processing multi-channel, high-frequency monitoring data, and has lower computational complexity, making it suitable for real-time monitoring tasks. Compared to the application of LSTM in long-sequence data, 1DCNN achieves effective data dimensionality reduction through convolution and pooling operations, reducing model complexity and computation. In contrast, while LSTM excels at capturing long-term dependencies, its larger number of parameters, longer training time on large-scale industrial data, and tendency to overfit limit its application in scenarios requiring high real-time performance.

The choice of 1DCNN as the monitoring model in this study is based on its superior performance in handling high-frequency, noisy industrial signals. 1DCNN not only quickly perceives the health status of equipment but also improves the model’s generalization capability and robustness against noise through the use of Dropout and pooling layers. Although LSTM performs well in certain scenarios, given the focus on high real-time performance and complex industrial settings in this study, 1DCNN is more appropriate. In future research, we will consider further comparing the performance of LSTM and 1DCNN to provide a more comprehensive performance analysis.

The relevant references have been added as follows:

(1) Mortezapour Shiri, Farhad & Perumal, Thinagaran & Mohamed, Raihani. (2023). A Comprehensive Overview and Comparative Analysis on Deep Learning Models: CNN, RNN, LSTM, GRU. 10.13140/RG.2.2.11938.81609.

(2) Li Hanlin, Niu Shaozhang, Wang Maosen, et al. Real-time Detection Method for Equipment Abnormal Response Based on Feature Mining [J]. Artificial Intelligence, 2023, (04): 71-81. DOI: 10.16453/j.2096-5036.2023.04.008.

(3) Sun Yan. A Brief Discussion on the Optimization of Convolutional Neural Networks [C] // Tianjin Electronics Society. Proceedings of the 37th China (Tianjin) 2023 IT, Network, Information Technology, Electronics, and Instrumentation Innovation Academic Conference. Tianjin Optoelectronic Communication Technology Co., Ltd.; 2023: 4.

---

## [Decision Letter · Decision Letter 2]

4 Oct 2024

Research on state perception of scraper conveyor based on one-dimensional convolutional neural network

PONE-D-24-05986R2

Dear Dr. Lu,

We’re pleased to inform you that your manuscript has been judged scientifically suitable for publication and will be formally accepted for publication once it meets all outstanding technical requirements.

Kind regards,

Brij Bhooshan Gupta

Academic Editor

PLOS ONE

Additional Editor Comments (optional):

Reviewers' comments:

Reviewer's Responses to Questions

**Comments to the Author**

1. If the authors have adequately addressed your comments raised in a previous round of review and you feel that this manuscript is now acceptable for publication, you may indicate that here to bypass the “Comments to the Author” section, enter your conflict of interest statement in the “Confidential to Editor” section, and submit your "Accept" recommendation.

Reviewer #1: All comments have been addressed

Reviewer #2: All comments have been addressed

2. Is the manuscript technically sound, and do the data support the conclusions?

Reviewer #1: Yes

Reviewer #2: Yes

3. Has the statistical analysis been performed appropriately and rigorously? 

Reviewer #1: N/A

Reviewer #2: Yes

4. Have the authors made all data underlying the findings in their manuscript fully available?

Reviewer #1: Yes

Reviewer #2: Yes

5. Is the manuscript presented in an intelligible fashion and written in standard English?

Reviewer #1: Yes

Reviewer #2: Yes

6. Review Comments to the Author

Reviewer #1: Well done you have incorporated all the suggestions. Kindly proof read for spelling and typo mistakes

Reviewer #2: The author's description of the advantages and disadvantages of 1DCNN and LSTM is not objective enough.

7. PLOS authors have the option to publish the peer review history of their article (what does this mean?). If published, this will include your full peer review and any attached files.

Reviewer #1: No

Reviewer #2: No

---

## [Editor Report · Acceptance letter]

8 Oct 2024

PONE-D-24-05986R2 

PLOS ONE

Dear Dr. Lu, 

I'm pleased to inform you that your manuscript has been deemed suitable for publication in PLOS ONE. Congratulations! Your manuscript is now being handed over to our production team.

Kind regards, 

on behalf of

Dr. Brij Bhooshan Gupta 

Academic Editor

PLOS ONE